# Pathogenic PSEN1 Glu184Gly Mutation in a Family from Thailand with Probable Autosomal Dominant Early Onset Alzheimer’s Disease

**DOI:** 10.3390/diagnostics10030135

**Published:** 2020-03-01

**Authors:** Vorapun Senanarong, Seong Soo A. An, Vo Van Giau, Chanin Limwongse, Eva Bagyinszky, SangYun Kim

**Affiliations:** 1Department of Medicine, Faculty of Medicine Siriraj Hospital, Mahidol University and Thailand, Bangkok 10700, Thailand; vorapun.sen@mahidol.ac.th (V.S.); siclw@mahidol.ac.th (C.L.); 2Department of Bionano Technology, Gachon University, Seongnam 13120, Korea; seong.an@gmail.com); 3Graduate School of Environment Department of Industrial and Environmental Engineering, Gachon University, Seongnam 13120, Korea; 4Department of Neurology, Seoul National University College of Medicine & Neurocognitive Behavior Center, Seoul National University Bundang Hospital, Seongnam 13620, Korea

**Keywords:** PSEN1, Alzheimer’s disease, mutation, presenilin1, next generation sequencing, splice site mutation

## Abstract

A pathogenic mutation in PSEN1 p.Glu184Gly was discovered in a Thai family with early onset Alzheimer’s disease (EOAD) as the first case in Asia. Proband patient presented memory impairment and anxiety at the age of 41 years. Family history was positive, since several family members were also diagnosed with dementia (father and grandfather). MRI in the patient revealed global cortical atrophy without specific lesions or lacuna infarctions. Extensive genetic profiling for 50 neurodegenerative disease related genes was performed by next generation sequencing (NGS) on the patient. PSEN1 Glu184Gly was previously reported in French families with frontal variant Alzheimer’s disease (AD). Interestingly, this mutation is located near the splicing site and could possibly result in abnormal cleavage of PSEN1 transcript. Furthermore, 3D models from protein structural predictions revealed significant structural changes, since glycine may result in increased flexibility of TM-III helix. Inter/intra-helical interactions could also be altered. In the future, functional studies should be performed to verify the probable role PSEN1 Glu184Gly in amyloid beta processing and pathogenicity.

## 1. Introduction

Alzheimer’s disease (AD) can be categorized into two forms according to age groups: early onset AD (EOAD) and late onset AD (LOAD), which may occur under and over 65 years of age, respectively. Percentages of patients with EOAD are relatively small, occurring in only a fraction (10–15%) of all AD cases [1,2]. Three genes were reported as causative factors for AD, accounting for less than 6% among EOAD cases [3]. They were the amyloid precursor protein (APP; NC_000021.9), presenilin 1 (PSEN1; NC_000014.9) and presenilin 2 (PSEN2; NC_000001.11). The majority of mutations in these three genes follow autosomal dominant inheritance patterns with the exception of autosomal recessive pattern (APP p.Glu693del and p.Ala673Val) [4,5,6]. Interestingly, one protective APP mutation was also reported, p.Ala673Thr (or Icelandic APP) [7]. Mutations in the above genes could be associated with impaired amyloid metabolism, resulting in higher Aβ42 and/or lower Aβ40 production or reduced clearances [8,9,10]. More than 200 pathogenic mutations for EOAD were reported in PSEN1 gene (AD and FTD mutation database; Alzforum database). PSEN1-associated AD may occur between 20 and 65 years of age. Several mutations in PSEN1, such as His163Pro [11], Leu226Phe [12,13], Leu232Pro [14], and Gly417Ala [15], were reported in relatively young onset AD patients, in which disease occurred under 40 years of age. The youngest juvenile AD patient (Leu85Pro) developed the disease phenotype when he was 27 years old [16]. Patients with young onset AD presented a very aggressive phenotype and rapid disease progression. Additional symptoms, such as Parkinsonism, epilepsy, and language and behavioral impairment may also be prominent in EOAD [17]. Interestingly, de novo cases of AD mutations were also reported without prior family history or any affected family members [2,18,19,20]. In this study, a pathogenic PSEN1 p.Glu184Gly (g.14:73659354 A>G; c.551A>G; p.E184G) mutation was discovered in an EOAD patient from a Thai family, noted as the first case in Asia. Structural characterization using 3D protein prediction was also carried out to understand the putative pathogenic mechanisms of this mutation [20]. 

## 2. Materials and Methods

Complex genetic investigation by next generation sequencing (NGS) was performed on the proband patient. Targeted NGS approaches were carried out with a panel of 50 causative and risk factor genes in neurodegenerative diseases by Theragen Etex Bio Institute on an IonProton device [20]. The standard Sanger sequencing verified the NGS data by BioNeer Inc. (http://eng.bioneer.com/home.aspx, Bioneer Inc., Daejeon, Korea). The father of patient was also tested for the pathogenic mutation by standard sequencing. 

## 3. Clinical Symptoms of Patient

A 41-year old female college lecturer complained about slowness of thoughts and forgetfulness, which appeared a year before her first visit to the hospital. Unremarkable neurological findings were seen upon examination of the patient, and her MMSE and MOCA scores were 29/30 and 25/30, respectively. Addenbrook IIIR score was 93/100. She performed poorly on Wechsler’s logical memory test (score less than 1.5SD of norms). Her visual memory, language, attention, and executive functions remained normal. One year after the first symptoms, she resigned from her university position and started working in her family business. The patient’s mother was pleased that the patient had quit from her job, since she seemed to be anxious and often forgot her teaching schedule. She was diagnosed with pure amnestic-mild cognitive impairment (MCI) after one year from her first symptoms. MRI scans revealed mild cortical atrophy, especially in the precuneus region. No white matter lesion, lacuna infarction, or cerebral microbleed were observed in the brain MRI (Figure 1). APOE genotype of patient was homozygous for E3 allele.

Patient (III-1) had a strong family history of early onset dementia (Figure 2). Her father had dementia in his early fifties and died at age 59 (II-2). Her two aunts ((II-5, II-6) were diagnosed with psychotic disorder, and even though they are still alive they became bedridden. The patient’s grandfather (I-2) had dementia in his mid-fifties and died at age 65. Her grandfather’s sister and brother (I-5, I-6) also had dementia in their mid-forties and both died at 60 years of age. After genetic counseling, the patient’s mother agreed to send a blood sample of the patient and her father to Seoul National University Bundang Hospital, for genetic analysis. However, patient and her family denied cerebrospinal fluid (CSF) and blood plasma analysis for AD biomarkers. All additional living family members refused the genetic test, or to give any additional information regarding their health. Written informed consent was obtained from the proband patient and her father for publication of this study. This report was approved by the Institutional Review Board of Seoul National University Bundang Hospital (B-1612/376-701). This study was approved in 2015 September

## 4. In Silico Analysis, Structure and Predictions

Variants were checked for their novelty in the Korean Genome Reference Database (http://152.99.75.168/KRGDB/menuPages/firstInfo.jsp), which provided the full genome sequences of 622 asymptomatic individuals by whole genome sequencing. In addition, variants were also monitored in larger genome reference databases, including the 1000Genomes (http://www.internationalgenome.org/) and Exome Aggregation Consortium (ExAC; http://exac.broadinstitute.org) databases. 

In silico analyses were performed by PolyPhen-2 (http://genetics.bwh.harvard.edu/pph2/), Sorting Intolerant From Tolerant (SIFT; http://sift.jcvi.org/) and PROVEAN (http://provean.jcvi.org/index.php) tools. They provided predictions on missense mutations, whether they were probable/possible pathogenic [21,22]. Additional protein function predictions were performed by the ExPASy (https://www.expasy.org/) tool with different parameters, such as bulkiness, polarity, or hydrophobicity (Kyte and Doolittle) indices. Splice site predication was also performed on the missense mutations using the Human Splicing Finder 3 (HSF3; http://www.umd.be/HSF3/) tool.

Protein structure of mutant and normal PSEN1 was predicted by Raptor X web software (http://raptorx.uchicago.edu/). For normal PSEN1, the crystal structure published by Bai et al. (2015) was used, which published the cryo-EM structure and atomic model of all γ-secretase components [23]. Discovery Studio 3.5 Visualizer by Accelrys Inc. was used for the comparison of the mutant and normal PSEN1 proteins [24].

## 5. Results

NGS and Sanger sequencing revealed a known pathogenic mutation in PSEN1 Glu184Gly (g.14:73659354 A>G; c.551A>G; p.E184G, Figure 3) [19] in the patient, and her father was also positive for the same mutation. No additional pathogenic mutation was found in other prominent causative genes of other neurodegenerative diseases, such as PD, FTD, ALS or prion disease (Appendix A). The mutation was missing in the KRGDB, ExAC and 1000Genomes databases, which confirmed that this report on PSEN1 Glu184Gly would be the first in Asian populations. In the same location (Glu184), another mutation in the form of Glu184Asp (Table 1) was reported in two Japanese families [25].

Multiple sequence alignment revealed that Glu184 was conserved among the majority of vertebrates (Figure 4). However, valine was found instead of glutamic acid at the homolog position of the PSEN-like sequence in toad species (Pipa carvalhoi). HumDiv and HumVar scores from PolyPhen2 analysis were 0.878 and 0.791, respectively, suggesting this mutation as possibly damaging. SIFT analysis also revealed Glu184Gly as a damaging variant with the score of 0.005. Again, PROVEAN analysis revealed Glu184Gly as damaging with the score of −6.110 (variants with scores of −2.5 or below may be predicted as damaging).

ExPASy tools, through different parameters, such as bulkiness, polarity, and hydrophobicity, revealed that the mutation could influence the presenilin structure significantly. Scores of bulkiness dropped significantly due to Glu184Gly (15.609) in comparison to normal PSEN1 (16.739), and several amino acids nearby were also affected starting from residues 180–189. PSEN1 Glu184Asp did not affect strongly the protein bulkiness (Figure 5a). Polarity scores were also reduced to 7.256 for the Glu184Gly mutation from the normal score of 7.622 in Figure 5b, while slight incensement in polarity was observed in case of Glu184Asp. In terms of hydrophobicity, increased scores were observed from the Glu184Gly mutation, while Glu184Asp did not change significantly (Figure 5c). These predictions suggested that glutamic acid and glycine have more distinct features than glutamic acid and aspartic acid and may result in more significant changes. However, ExPASY simulations may not be fully relevant to disease pathology, since both Glu184Gly and Glu184Asp could result in early onset form of disease. 

Comparing the Glu184Gly mutant and normal protein by structure predictions revealed significant changes in the TM helix, and in amino acid interactions with neighboring residues, especially with Val185 (Figure 6 a,b). PSEN1 Glu184Gly was located in the lumen or extracellular side of TM-III. Glycine substitution may induce the perturbation in the helix integrity, resulting in abnormal torsional motions. In addition, glycine may result in kinks inside the helix, which would be absent in the normal presenilin-1 protein. The exchange of negatively charged glutamic acid to the hydrophobic glycine could also alter the intra-helical hydrophobic interactions and affect the electrostatic interactions with neighboring amino acids. Glu184 may interact with several amino acids inside the helix, including Ile180, Tyr181, Phe186, and Thr188. In addition, the predicted PSEN1 structure with Gly184 revealed several altered hydrophobic interactions with Leu182, Phe186, and Thr188, leaving only interactions with Ile180 and Tyr181 (Figure 6a,b). Glu184Gly showed a higher degree of changes in the protein structure and hydrophobic interactions in comparison to the structure PSEN1 Glu184Asp. The smaller Asp184 may result in milder changes in helix flexibility and hydrophobic interactions [24] (Figure 6c).

Since Glu184Gly is located near the 5′ end of exon 7, HSF3 tool was used to perform the splice site prediction. Three potential acceptor sites (−6 tttcagGGAAGTG, −11 ttttttttcagGGA; −12; attttttttcagGG) were predicted to locate near this region. In addition, another potential acceptor site and two potential donor sites were also found near the mutation (−3; cagGGAAGT and +1; GAAGTGTTT (Figure 7a)). Mutation may eliminate the +2; GAAGTGTTT donor and the −11 ttttttttcagGGA acceptor sites. However, PSEN1 Glu184Gly may also impact the other splice sites nearby (Figure 7b). However, mRNA gene expression study was not carried out to verify the alterations in the splicing product. 

## 6. Discussion

In this study, a PSEN1 Glu184Gly mutation was reported in a 41-year old EOAD patient with strong family history of early onset dementia from Thailand. Previously, this mutation was reported from French EOAD patients with autosomal dominant pattern (Table 1), and the ages of onset were between 43–52 years with a disease duration of 5 to 14 years. These patients presented two atypical phenotypes as follows: frontal variant of AD (behavioral impairment) and early myoclonic epilepsy. Unfortunately, no detailed report was available on the clinical and neuropathological phenotypes of the French patients [19]. The Thai patient family was also associated with familial case of disease. The patient’s father presented with memory complaint, while the patient’s aunts were diagnosed with psychosis. Age of onset and disease duration was similar in the affected members of Thai and French families [19].

A second mutation for the same residue, Glu184Asp (Table 1), was discovered in two Japanese families. In the first family, the patient experienced amnesia and disorientation in his early 40s. Mild intellectual impairment was followed by severe memory dysfunction, rigidity in the muscles, seizures, and mild extrapyramidal signs. Family history was strong: the patient’s sister also developed tremor and dementia in her 40s, and similar symptoms were observed in his mother and grandmother [24,25,26]. In the second Japanese family, a patient developed dementia with Lewy bodies (DLB)-like phenotype with hallucinations, delusions, and parkinsonism. Neurofibrillary tangles, senile plaques, and amyloid angiopathy were present in the brain of the affected individuals in both families. In the patient of second family, α-synuclein accumulation also occurred in the plaques. This study suggested that either Lewy body pathology could influence the amyloid processing, or Glu184Asp may play a role in Lewy body formation through some unidentified pathway [24,25,26]. PSEN1 Glu184Asp was also identified in a Caucasian family from the UK, where the age of onset occurred between 40–44 years [27]. In addition, mutation was found in a Czech patient, who developed familial EOAD at the age of 44. She presented primary progressive aphasia, language and behavioral dysfunctions, and episodic memory impairment. Neuropathological examination confirmed the diagnosis of EOAD with Lewy bodies, but metastatic tumor also appeared in the parietal lobe from her colon cancer [27].

Since the functional test or CSF triple biomarkers were not available, several bioinformatics methods were used to assess the significance of PSEN1 Glu184Gly in disease progression. All PolyPhen2, SIFT and PROVEAN tools revealed the mutation as a damaging variant. ExPASy tools showed that PSEN1 Glu184Gly in the extracellular part of TM-III of PSEN1 could affect the protein structure and function through changes in bulkiness, polarity, and hydrophobicity. From the protein structure prediction, this mutation could disturb the protein structure due to the flexibility of glycine. Glycine could appear in loop regions and beta-turns in the soluble proteins, and was suggested as a helix-breaker in TM helices. Glycine residues in helices could be also involved in protein interactions by stabilizing the membrane structure [28,29]. An extra glycine residue in the α-helices may result in altered torsional motions. Abnormal motion of PSEN1 TM-III may impact the γ-secretase activity or assembly. PSEN1 Glu184Gly may also interfere with the intramolecular interactions, and the connection between TM-III and the other transmembrane domains/PSEN1 binding partners. 

PSEN1 Glu184Gly is located near the 5′ end of PSEN1 exon 7 and could affect the splicing of PSEN1 transcript by disturbing the natural acceptor and possible acceptor and donor sites. Mutations located near the end of exons could possibly affect the splicing of PSEN1. Several mutations were verified to impact the PSEN1 splicing. For example, Leu113insThr resulted in partial or full deletion of exon 4 [30]. PSEN1 Gly183Val revealed in skipping exon 6 or both exon 6–7, resulting in reduced PSEN1 function [31]. PSEN1 Ser290Cys was located at the splice junction of exon 8 and 10, causing deletion of entire exon 9 [32,33]. Additional mutations indicated significant impacts on PSEN1 mRNA splicing, such as Leu113Pro [34], Leu113Gln [35], Glu184Gly [19], Glu184Asp [24], and Arg377Trp [19]. Mutations, eliminating the splice acceptor or donor sites, could result in abnormal cleavage of exons. Skipping exons produced non-functional PSEN1 protein and dysfunctions in γ-secretase cleavage [30]. 

PSEN1 protein is composed of nine non-polar TM domains (TM), connected with hydrophilic (HL) loops [23,36]. The C-terminal TM domain in the APP protein could interact with the PSEN1 TM domains (TM-II, TM-IV). Several mutation hotspots were described in PSEN1 TM-II and TM-III, such as Met139, Ile143, Met146, His163, Leu166, Ser169, or Leu177. Mutations in these regions may result in conformational changes in γ-secretase complex, resulting in altering the interactions between the enzyme and the APP protein. The other putative mechanism could be that these mutations may prevent some essential conformational changes for the proper enzyme-substrate interactions [37]. TM-III could affect the motion of TM-II, resulting in “semi-open” or “innocent” conformation of γ-secretase. These two domains may play a role in controlling the availability of active site of enzyme. “Semi-open” stage could be an optimal conformation for APP cleavage, producing short Aβ peptides (Aβ40 or Aβ38). PSEN1 mutations may induce the “open” conformation of γ-secretase, resulting in increased ratio of long/short amyloid peptides. It may be possible that PSEN1 Glu184Gly could also disturb the semi-open form of γ-secretase and result in elevated long Aβ peptide (Aβ42) production [38]. Mutations in PSEN1 and PSEN2 may perturb the stability of γ-secretase complex and affect the amyloid production [8,9]. Amyloid peptide species may not be limited to Aβ42 and longer amyloid peptides such as Aβ43, Aβ48, and Aβ49, which revealed higher aggregation potentials and result in increased neurotoxicity [10]. Numerous mutations in PSEN1 could increase the levels of longer amyloid peptides in comparison to the shorter ones (Aβ49 > Aβ40 and Aβ48 > Aβ38), which could be due to the alteration of active site and of γ-secretase complex. AD causative mutations could also shift the site of initial ε-cleavage (endoproteolysis), resulting in the release of premature Aβ43 or Aβ42 [8,10].

In vitro studies were performed by Shi et al. (2017). PSEN1 Glu184Gly mutation elevated the Aβ42/Aβ40 ratio, compared to wild type Neuro2a (N2a) cell lines [39]. However, further studies may be needed on this mutation to verify the disease mechanisms, associated with PSEN1 Glu184Gly. Several mutations were found in TM-III, which resulted in elevated Aβ42 levels or Aβ42/Aβ40 ratio. 

One of the limitations of this study of this PSEN1 Glu184Gly variant was the absence of CSF biomarker studies. In addition, relatives of patient refused the genetic test for PSEN1 Glu184Gly. Furthermore, acquiring brain tissue from the patient to examine the abnormal splicing of PSEN1 was impossible due to ethical reasons in Thailand. 

This report is the first PSEN1 Glu184Gly on an Asian family, following the two French cases. This mutation was confirmed as a causative factor for EOAD, since all patients with PSEN1 Glu184Gly had positive family history for EOAD, and the mutation was missing in reference databases [19,40]. Bioinformatics studies also strongly characterized this variant as pathogenic. In silico structure predictions of this mutation revealed significant disruption in TM-III and altered the helix structure. The mutation may also result in alternative splicing of PSEN1 mRNA. The codon 184 would be an important hotspot in PSEN1, since age of onset was found to be as early as 40 years for EOAD for both Glu184Gly and Glu184Asp.

## Figures and Tables

**Figure 1 diagnostics-10-00135-f001:**
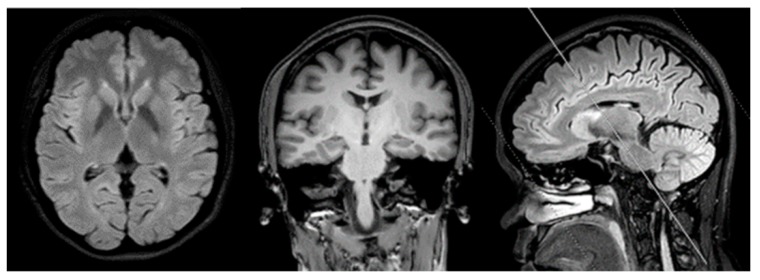
Magnetic resonance imaging of the brain (1.5T Philips) of the patient. No white matter lesion or lacunar infraction No medial temporal atrophy was detected. Mild global cortical atrophy for her age was detected.

**Figure 2 diagnostics-10-00135-f002:**
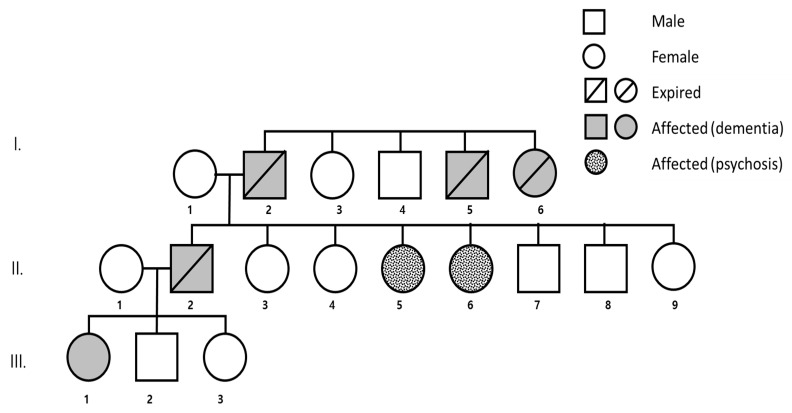
Family tree of patient (III-1) with PSEN1 E184G. This mutation was associated with a strong family history of dementia.

**Figure 3 diagnostics-10-00135-f003:**
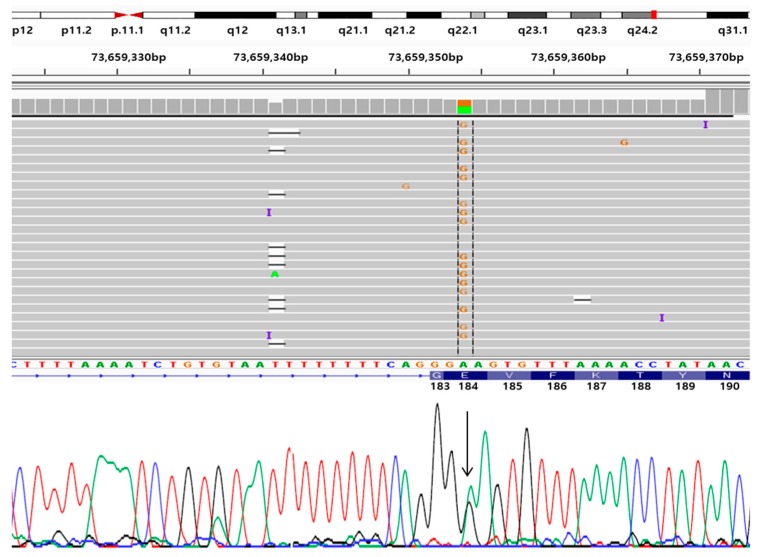
Next generation sequencing (NGS) data, aligned with standard sequencing data for PSEN1 Glu184Gly mutation. (G means guanine, C means cytosine T means Thymine, A means adenine)

**Figure 4 diagnostics-10-00135-f004:**
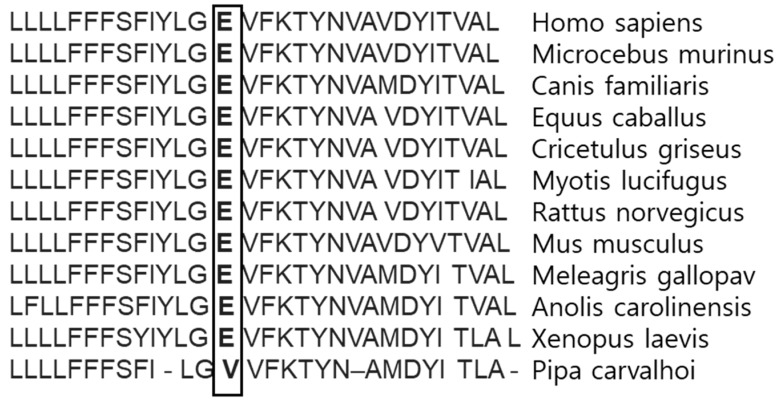
Multiple sequence alignment on human PSEN1 protein sequence, compared to different vertebrate species. Glu184 seems to be conservative among the majority of vertebrate species. Glu184 was highlighted with bold in the different species.

**Figure 5 diagnostics-10-00135-f005:**
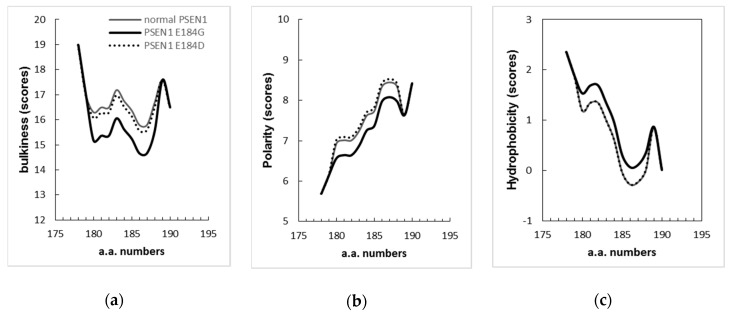
ExPASy predictions for PSEN1 Glu184Gly, compared to normal PSEN1 and Glu184Asp structure: (**a**). in terms of bulkiness; (**b**). in terms of polarity; (**c**). in terms of Kyte & Doolittle hydrophobicity index. Scores of bulkiness and polarity have been dropped significantly due to the mutation, while the hydrophobicity scores have been increased because of the mutation. “a.a. numbers” mean amino acid numbers.

**Figure 6 diagnostics-10-00135-f006:**
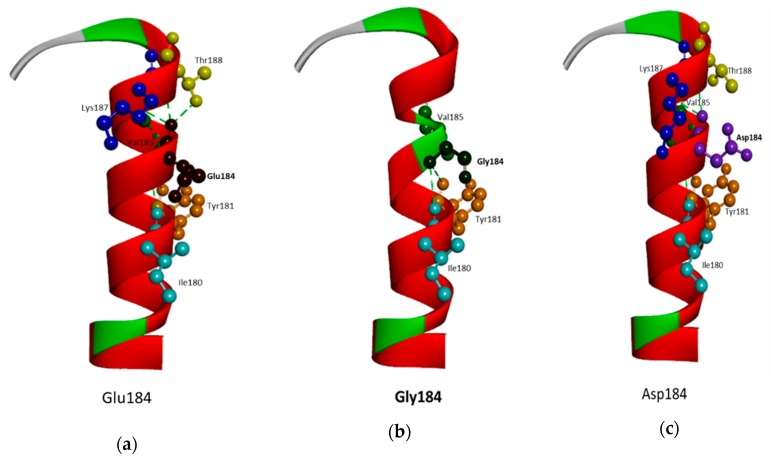
3D structure prediction for PSEN1 Glu184Gly and Glu184Asp, comparing the normal PSEN1 with the mutant PSEN1. A. Location of Glu184 in PSEN1 protein (C-terminal end of TM-III, in cytosol). B. In the normal PSEN1, Glu184 is in hydrophobic interaction in several molecules, including Thr188, Lys187, Tyr181, and Ile180. C. Due to Gly184, most of hydrophobic contact may be lost, except Tyr180 and Ile181. In addition, the mutation may result in a kink-like structure, due to the helix breaker properties of glycine. Asp184 may result in less significant change in protein structure, as well as hydrophobic interactions, than Gly184.

**Figure 7 diagnostics-10-00135-f007:**
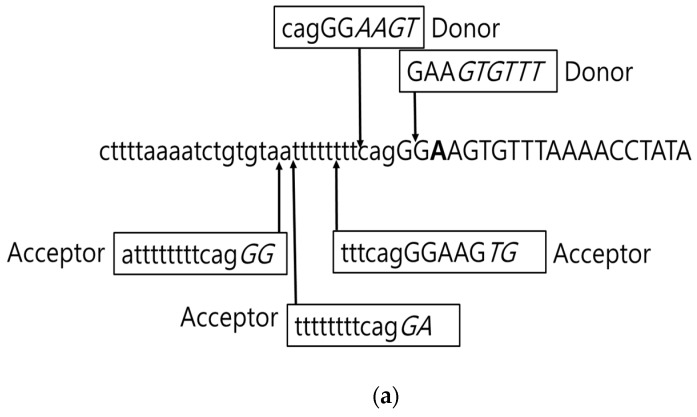
Splice site prediction in PSEN1 Glu184Gly, predicted by HSF 3 tool. a. Potential splice sites in the normal PSEN1 sequence. b. Potential splice sites in case of PSEN1 Glu184Gly mutation.

**Table 1 diagnostics-10-00135-t001:** Comparisons of dominant features of the patients with PSEN1 E184G and with E184D.

Case	E184G	E184G (Our Case)	E184D
**Age of first symptoms**	43–52 years	41–52 years	40–44
**Familiy history**	Positive	Positive	Positive
**Main symptoms**	Frontal variant of AD Myoclonic epilepsy neuritic plaques in the frontal cortex, severe cotton wool plaques and Lewy bodies pathology.	Memory impairment, slowness in thought at age of 41. Her father had dementia in early fifties and died at 59. The father’s sisters also had dementia presenting with psychosis in their late forties. Grandfather also had dementia in his fifties and died at 60.	Memory impairment, tremor, rigidity in the muscles, seizures and mild extrapyramidial signs. Language impairment. Dementia with Lewy bodies (DLB)-like phenotype was also associated with E184D.
**Disease duration**	5–14 and 7–8 years	8–10 years	7–10 years
**Diagnosis at** **Autopsy**	Compatible with AD.	Autopsy of the patient’s father was not done.	Compatible with AD.
**Nationality of patients**	France	Thailand	UK, Czech Republic and Japan
**Reference**	[19].	Current study	[24,25,26].

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
