# Peer review of "Pathogenic PSEN1 Glu184Gly Mutation in a Family from Thailand with Probable Autosomal Dominant Early Onset Alzheimer’s Disease"

_diagnostics, 2020, doi:10.3390/diagnostics10030135_

Round 1

Reviewer 1 Report

Please review for grammar and overall flow as transitions are not as smooth as they could be.

Author Response

Thank you very much, English improvement was performed on manuscript.

Reviewer 2 Report

Senanarong et al. present somewhat of a follow-up manuscript to their 2019 work entitled, “Analysis of 50 Neurodegenerative Genes in Clinically Diagnosed Early-Onset Alzheimer’s Disease.” In the manuscript under review they provide a more detailed description of an early-onset Alzheimer’s disease patient that they first identified in their screen of a Thai cohort of similar patients. The patient has a highly-likely pathogenic mutation in the gene encoding presenilin 1 (PSEN1). The authors provide a family history, clinical details of the proband, and in silico profiling of the Glu184Gly mutation. This case study is interesting and of use to the field. I have a few minor suggestions that the authors should consider:

I find it rather odd that the authors do not discuss and cite their aforementioned previous study (Int. J. Mol. Sci. 2019, 20, 1514; doi:10.3390/ijms20061514). I think the authors should include reference to this work to set the scene better for this current manuscript. Please provide references for the final sentence of the Introduction (I appreciate they are included later, but they would be helpful here too). So that readers do not have to go to the previous paper, it would be useful to have the list of 50 genes on the panel that were screened for – please provide (supplementary?) An evolutionary alignment of PSEN1 is mentioned, yet not included. A visual representation would be helpful. Given that hydrophobicity, bulkiness and polarity are all largely unaffected in the pathogenic E184D PSEN1 mutant, it suggests that their alterations in the E184G mutant are perhaps not particularly relevant to pathology – can the authors please comment on this? Figure 6 is hard to interpret. Please improve both the legend and the figure for clarity. In the table, the “Age of first symptoms” is stated as 41-44, but in the text it states 40-44; please clarify.

Author Response

I find it rather odd that the authors do not discuss and cite their aforementioned previous study (Int. J. Mol. Sci. 2019, 20, 1514; doi:10.3390/ijms20061514). I think the authors should include reference to this work to set the scene better for this current manuscript. Please provide references for the final sentence of the Introduction (I appreciate they are included later, but they would be helpful here too).

Thank you, reference has been added to manuscript.

 So that readers do not have to go to the previous paper, it would be useful to have the list of 50 genes on the panel that were screened for – please provide (supplementary?).

Thank you, supplementary table was added on variants, found in patient.

 An evolutionary alignment of PSEN1 is mentioned, yet not included. A visual representation would be helpful.

Thank you, figure was added (Figure 4.).

"Figure 4. Multiple sequence alignment on human PSEN1 protein sequence, compared to different vertebrate species. Glu184 seems to be conservative among majority of vertebrate species."

Figure 6 is hard to interpret. Please improve both the legend and the figure for clarity.

Thank you, this issue has been fixed. We redrew the figure, and made separate figure on splice site predictions on normal and mutant PSEN1 exon 7.

 "Figure 7. Splice site prediction in PSEN1 Glu184Gly, predicted by HSF 3 tool. a. Potential splice sites in the normal PSEN1 sequence b. Potential splice sites in case of PSEN1 Glu184Gly mutation"

“Since Glu184Gly is located near the 5’ end of exon 7, HSF3 tool was used to perform the splice site prediction. Three potential acceptor sites (-6 tttcagGGAAGTG,-11 ttttttttcagGGA; -12; attttttttcagGG ) were predicted to locate near this region. In addition, another potential acceptor site and two potential donor sites were also found near the mutation (-3; cagGGAAGT and +1; GAAGTGTTT; Figure 7.a). Mutation may eliminate the “+2; GAAGTGTTT” donor and the ,-11 ttttttttcagGGA acceptor sites, however, PSEN1 Glu184Gly may also impact the other splice sites nearby. (Figure 7.b.).  However, mRNA gene expression study was not carried out to verify the alterations in the splicing product.”

In the table, the “Age of first symptoms” is stated as 41-44, but in the text it states 40-44; please clarify.

Thank you, issue has been fixed.